

# Antimicrobial resistance pattern of *Escherichia coli* isolated from imported frozen shrimp in Saudi Arabia

Ibrahim Alhabib and Nasreldin Elhadi

Department of Clinical Laboratory Sciences, College of Applied Medical Sciences, Imam Abdulrahman Bin Faisal University, Dammam, Eastern Region, Saudi Arabia

## ABSTRACT

Contamination of seafood products with multi-drug-resistant (MDR) bacteria is considered to be a potential source for the spread of MDR bacteria in communities. However, little is known about the extent of the contamination of seafood, in particular shrimp, with MDR bacteria in Saudi Arabia. In this study, imported frozen shrimp in retail markets were examined for the antimicrobial susceptibility patterns of *Escherichia coli*. *Escherichia coli* was isolated from 40 frozen shrimp samples; a total of 25 and 15 shrimp samples were imported from China and Vietnam, respectively. Of the 40 examined frozen shrimp samples, 30 tested positive for *E. coli*, resulting in an overall isolation rate of 75%, with a total of 180 isolates being identified. The largest number of positive samples for *E. coli* isolates ($n = 140$) were found in 22 out of 25 samples from frozen shrimp imported from China. Additionally, eight out of 15 samples from frozen shrimp imported from Vietnam were positive for *E. coli*, leading to the recovery of 40 isolates. Overall, the susceptibilities among *E. coli* isolates were observed against 99.4% for amikacin, augmentin and kanamycin, 95% for cefoxitin and 92.7% for ceftazidime and nitrofurantoin. High percentage of the isolates exhibited resistance to cephalothin (174/180, 96.6%), ampicillin (167/180, 92.7%), Cephalexin (163/180. 90.5%), piperacillin (156/180, 86.6%), ceftriaxone (123/180, 68.3%), nalidixic acid (95/180, 52.7%), trimethoprim sulphamethoxazole (90/180, 50%), and tetracycline (88/180, 48.8%). Additionally, the MDR (resistant to ≥3 different class of antimicrobials) rate among *E. coli* isolates was 94.4% (170/180). A relatively high intermediate resistance of 60% (108/180) was exhibited for aztreonam. Notably, *E. coli* showed 71 different antibiotic resistance patterns with the multiple antibiotic resistant (MAR) index ranging from 0.04 to 0.66 and 89.4% of *E. coli* isolates recorded very significant MAR indexes above the range of 0.2. In this study, we recorded highest antimicrobial co-resistance patterns of 162 (90%) isolates between cephalothin and ampicillin, followed by 158 (87.7%) isolates between cephalothin and cephalexin. Furthermore, and interestingly, the segregation of antimicrobial resistance patterns based on the source of country origin of frozen shrimp revealed few inter-country resistant patterns found interconnecting and this influenced 44 (24.4%) isolates overlapping between isolates origin from frozen shrimp imported from China and Vietnam. This study documents the antimicrobial resistance in *E. coli* isolated from imported frozen shrimp and the presence of MDR *E. coli* in shrimp consuming communities, which

Corresponding author
Nasreldin Elhadi,
nmohammed@iau.edu.sa

may pose a risk to public health and the transfer of resistant bacteria to the food chain and environment.

## INTRODUCTION

*Escherichia coli (E. coli)*, a member of the *Enterobacteriaceae* family, is known to be a foodborne pathogen in humans (*Skurnik et al., 2006*; *Addis & Sisay, 2015*; *Davis et al., 2018*). It is able to spread along the food chain and into ecosystems and has been widely reported worldwide. *E. coli* may be of pathogenic or non-pathogenic strains, causing disease in both the intestinal tract and other areas of the body. The main conditions or clinical signs of an *E. coli* infection can be observed as patients having diarrhea, urinary tract infections, meningitis, peritonitis, septicemia and gram-negative bacterial pneumonia. (*Hammerum & Heuer, 2009*). Strains of *E. coli* of animal origin can be opportunistic and pathogenic, which may lead to either less harmful infections (*e.g.*, uncomplicated urinary tract infections) or lethal infections (*e.g.*, blood stream infections) even though the fact that the majority of *E. coli* strains are non-pathogenic or even part of the normal flora in humans (*Hammerum & Heuer, 2009*).

Seafood is consumed globally, and the probability of ingesting seafood with pathogenic microorganisms may be observed in unsanitary conditions either *via* the handling and storage processes or exposure to contaminated water sources. (*Feldhusen, 2000*). *Escherichia coli* is one of the pathogenic organisms that are considered emergent in aquaculture, which is the fastest growing food producing sector according to the United Nations Food and Agriculture Organization (FAO) (*Barbosa et al., 2016*). Moreover, consumption of raw or undercooked seafood will increase chances of *E. coli* outbreaks emphasizing the importance of spreading public health awareness to these potential health risks with subsequent diseases that may occur through contamination (*Buchanan & Doyle, 1997*; *Barbosa et al., 2016*). Furthermore, the public should be informed that some *E. coli* strains are heat resistant and may spread infections even when precautions have been made to thoroughly cook food (*Li & Gänzle, 2016*).

Antimicrobial resistance (AMR) is another key factor when discussing the impact of *E. coli* infection on humans. This is supported by the fact that AMR is considered a global threat to the public health and economy (*Baekkeskov et al., 2020*). Additionally, the use of antibiotics in food-producing animals is increasing despite the known implications to human health (*Van Boeckel et al., 2015*; *Davis et al., 2018*). Furthermore, it was demonstrated that AMR could be transferred between commensal and zoonotically pathogenic members of the *Enterobacteriaceae* through the transmission of genetic material (*Blake et al., 2003*). Hydrolysis of β-lactam antibiotics *via* overproduction of β-lactamases is one of the most commonly reported resistance mechanisms of *Enterobacteriaceae* (*De Angelis et al., 2020*). Extended-spectrum β-lactamases (ESBLs)

from microbes, in particular *E. coli*, are variant of β-lactamases that confer resistance to multiple drugs including aztreonam, cefotaxime, ceftazidime, and related oxyimino-β-lactams as well as to other penicillin and cephalosporins. Although they may be inhibited by β-lactamases inhibitors such as clavulanic acid and tazobactam (*Pitout et al., 1998*; *Fernandes et al., 2014*). In addition to the possibility of transferring multidrug-resistance capacity between multidrug-resistant microbe, ESBL producing *E. coli* are of particular importance when considering the potential health issues worldwide (*Bonnet, 2004*).

In 2019, the estimated annual mortality rate attributed to AMR was approximately 1.27 million deaths, with developing countries bearing the greatest impact of this burden (*Murray et al., 2022*). AMR from *E. coli* is no longer through healthcare associated infections (nosocomial outbreaks), as reports have shown that both humans and animals contribute to the contamination of aquatic systems being significant reservoirs of resistance to antibiotics. Aquatic environments are emerging as significant reservoirs of antibiotic resistance, enabling the spread of antibiotic resistance genes (ARGs) through contaminated food sources (*Nnadozie & Odume, 2019*; *Taneja & Sharma, 2019*; *Koutsoumanis et al., 2021*). The inappropriate use of antibiotics in aquaculture significantly contributes to the global spread of AMR which has resulted in severe repercussions which has impacted human, animal and environmental health (*Preena et al., 2020*; *Caputo et al., 2023*). A concerning direct correlation between AMR and ARGs was shown when bacterial strains were found in seafood sources leading to more recent studies highlighting that most of these strains were from *E. coli* that carry β-lactam antibiotic resistance genes (*Mughini-Gras et al., 2019*; *Celik et al., 2023*). With the a growing global population that is increasingly dependent on aquaculture products for their food security, the rise of AMR and related infections linked to seafood poses a significant threat to public health (*Thornber et al., 2020*). Although the aquaculture supply chain plays a crucial role and is recognized within the context of a "One Health" framework for controlling the spread of AMR in the global aquaculture sector (*Lee & Brumme, 2013*; *Caputo et al., 2023*). In 2015, during the 68[th] World Health Assembly held in Switzerland and subsequent to the adoption of the Global Action Plan (GAP) on Antimicrobial Resistance (AMR), most members of the United Nations (UN) and World Health Organization (WHO) implemented a resolution that committed to the development of National Action Plans (NAPs) based on a "One Health" approach to reduce the use of antibiotics in hope to decrease the spread of antimicrobial resistance (AMR) (*Food and Agriculture Organization of the United Nations, 2016*).

A significant portion of food necessities of Saudi Arabia is maintained through imports from different countries worldwide (*El Sheikha, 2015*; *Elzaki & Al-Mahish, 2024*). Food products imports including seafood are regulated by the Saudi Food and Drug Authority (SFDA), which require a health certificate for imported seafood of animal origin among other conditions and requirements ensuring food safety (*Saudi Food and Drug Authority, 2021*). Despite the local and international strict regulations to ensure food safety, the ability of microorganisms to modify *via* mutations so as to acclimate efficiently towards the new surroundings suggest the requirement for continuous development of the regulations and monitoring strategies to aid in the prevention and management of food-related diseases

(*Lorenzo et al., 2018*). Contamination of seafood products with antibiotic-resistant bacteria is a growing concern and poses a serious public health issue. This study aimed to investigate the antimicrobial resistance as well as assess the extent of co-resistance patterns in *E. coli* isolated from imported frozen shrimp available for purchase in the Eastern Province of Saudi Arabia.

## MATERIALS AND METHODS

### Samples

In this study, a total of 40 samples of frozen shrimp imported from China ($n = 25$ samples) and from Vietnam ($n = 15$ samples) were purchased from different supermarkets in Al Khobar in the Eastern Province of Saudi Arabia. Each purchased sample of shrimp is equivalent to one kilogram. All samples were examined for the presence of multidrug-resistant (MDR) *E. coli*. During sample processing, the storage temperature and production date of the products were noted. Most of the purchased frozen shrimp in this project were peeled shrimp without head and some samples were found without tails. The purchased samples were transferred to the microbiology laboratory and were stored at $-20\ ^\circ$C.

### Isolation and identification

*Escherichia coli* enrichment broth (EC Broth, Oxoid, Hampshire, England) and CHROMagar™ *E. coli* (CHROMagar, Saint-Denis, France) were used to isolate *E. coli* from frozen shrimp. Briefly, a 25 g portion of shrimp from each sample was weighed and placed in a sterile stomacher plastic bag containing 225 mL of EC Broth (*Elhadi et al., 2004*). The bag was placed into the stomacher 400 lab blender (Stomacher 400 Circulator; Seward, West Sussex, UK) and blended for 2 min. Then, the broth containing the sample was incubated at 37 °C for 24 h. After incubation, 10 μL from each broth were taken by an inoculating loop and streaked onto CHROMagar™ *E. coli* plates and were incubated at 37 °C. After 24 h of incubation plates were inspected for blue-colored colony formation and were identified using biochemicals which include oxidase, indole and API 20E kit strips (BioMerieux, Marcy, France).

### Antibiotic susceptibility testing

Antimicrobial susceptibility was assessed in accordance with the established protocol for the Kirby-Bauer disk diffusion susceptibility test. For each test suspension the turbidity was adjusted to a 0.5 MacFarland and inoculated onto Muller-Hinton Agar using sterile cotton-wool swabs. A total of twenty-one different Oxoid antibiotic discs (Oxoid, Hampshire, UK) from different classes were tested. The antibiotic discs were dispensed onto the agar using an automated disk dispenser (Oxoid, Hampshire, UK). The diameter of the inhibition zone was measured in millimeters with Vernier calipers, and the results were interpreted as sensitive (S), intermediate (I), or resistant (R) based on breakpoints for *E. coli* as per the protocol established by *Hudzicki (2009)* and the Clinical and Laboratory Standards Institute (CLSI) (*CLSI, 2019*). The tested antibiotic agents included: ampicillin (AM, 10 μg), amikacin (AK, 30 μg), augmentin (AUG, 30 μg), aztreonam (ATM, 30 μg),

ciprofloxacin (CIP, 5 µg), cefotaxime (CTX, 30 µg), ceftazidime (CAZ, 30 µg), ceftriaxone (CRO, 30 µg), chloramphenicol (C, 30 µg), cephalexin (CFX, 30 µg), cefoxitin (FOX, 30 µg), nitrofurantoin (FM, 50 µg), gentamicin (GM, 10 µg), kanamycin (K, 30 µg), cephalotin (KF, 30 µg), nalidixic acid (NA, 30 µg), norfloxacin (NOR, 10 µg), tobramycin (TN, 30 µg), piperacillin (PIP, 30 µg), trimetrophrim sulphamethoxazole (SXT, 1.25 µg/23.75 µg) and tetracycline (TE, 30 µg). The *Escherichia coli* ATCC25922 reference strain included with the samples served as the control in performed tests. Determining the MAR index was performed using *E. coli* isolates following the Krumperman method (*Krumperman, 1983*). The MAR index was calculated by utilizing the equation MAR = a/b, with "a" denoting the count of antibiotics to which isolates showed resistance, and "b" representing the total number of antibiotics used in this study. More than or above 0.2 in valuesuggests that the isolates were obtained from high-risk origins. MDRwas characterized as being resistant to ≥3 different classes of antimicrobials (*Magiorakos et al., 2012*).

# RESULTS

## Isolation and prevalence rate

The analysis of 40 frozen imported shrimp samples, 30 tested positive for *E. coli*, resulting in an overall prevalence rate of 75%, with a total of 180 isolates being identified. The largest number of *E. coli* isolates ($n = 140$) were found in 22 (88%) out of 25 positive samples from frozen shrimp imported from China. Additionally, eight (53.3%) out of 15 samples from frozen shrimp imported from Vietnam were positive for *E. coli*, leading to the recovery of 40 isolates (Table S1).

## Antibiotic susceptibility testing

All the 180 *E.coli* strains isolated from imported frozen shrimp were subject to antibiotic susceptibility testing with a battery of 21 distinct antibiotics. The antibiotic susceptibility results are shown in Table 1. Overall, the highest percentage of isolates exhibited resistance to cephalothin 174 (96.6%), ampicillin 167 (92.7%), cephalexin 163 (90.5%), piperacillin 156 (86.6%), ceftriaxone 123 (68.3%), nalidixic acid 95 (52.7%), trimethoprim-Sulphamethoxazole 90 (50%), and tetracycline 88 (48.8%), respectively (Fig. 1). Relatively, higher susceptibilities were observed against augmentin 179 (99.4%), amikacin 179 (99.4%); kanamycin 179 (99.4%); cefoxitin 171 (95%); ceftazidime 167 (92.7%); and nitrofurantoin 167 (92.7%) as shown in Table 1. Surprisingly, results for aztreonam, from 108 isolates were found to be intermediate resistant marking up to 60.0% (Fig. 1). The lowest resistance of 0.5% was found similar to the subsequent antibiotics amikacin, augmentin and kanamycin (Fig. 1).

## Antibiotic resistance and multiple drug resistance patterns

The multiple antibiotic resistance patterns and the multiple antibiotic resistance (MAR) index are presented in Table 2. Among the total of 180 isolates of *E. coli* isolated from frozen shrimp imported from China and Vietnam, none of the isolates were found susceptible to any of the antibiotics used (Table 2). The highest number of isolates 40 (22.2%) exhibited resistance to eight antimicrobials, with 33 (18.3%) found in frozen

Table 1 Overall antibiotic susceptibility testing of *E. coli* (China: *n* = 140 and Vietnam: *n* = 40) isolated from imported frozen shrimp based on country of origin.

| Antibiotic class | Antibiotic | Resistance *n* (%) | | | Intermediate *n* (%) | | | Susceptible *n* (%) | | |
|---|---|---|---|---|---|---|---|---|---|---|
| | | China | Vietnam | Total | China | Vietnam | Total | China | Vietnam | Total |
| Cephalosporins | Ciprofloxacin (CIP, 5 µg) | 76 (54.2) | 1 (2.5) | 77 (42.7) | 1(0.7) | 2 (5) | 3 (1.6) | 63 (45) | 37 (92.5) | 100 (55.5) |
| | Cefotaxime (CTX, 30 µg) | 44 (31.4) | 27 (67.5) | 71 (39.4) | 61 (43.5) | 6 (15) | 67 (37.2) | 35 (25) | 7 (17.5) | 42 (23.3) |
| | Ceftazidime (CAZ, 30 µg) | 10 (7.1) | 0 | 10 (5.5) | 3 (2.1) | 0 | 3 (1.6) | 127 (90.7) | 40 (100) | 167 (92.7) |
| | Ceftriaxone (CRO, 30 µg) | 97 (69.2) | 26 (65) | 123 (68.3) | 20 (14.2) | 9 (22.5) | 29 (16.1) | 23 (16.4) | 5 (12.5) | 28 (15.5) |
| | Cephalexin (CFX, 30 µg) | 126 (90) | 37 (92.5) | 163 (90.5) | 1 (0.7) | 2 (5) | 3 (1.6) | 13 (9.2) | 1 (2.5) | 14 (7.7) |
| | Cefoxitin (FOX, 30 µg) | 8 (5.7) | 0 | 8 (4.4) | 1 (0.7) | 0 | 1 (0.5) | 131 (93.5) | 40 (100) | 171 (95) |
| | Cephalothin (KF, 30 µg) | 137 (97.8) | 37 (92.5) | 174 (96.6) | 0 | 0 | 0 | 3 (2.1) | 3 (7.5) | 6 (3.3) |
| Penicillins | Ampicillin (AM, 10 µg) | 127 (90.7) | 40 (100) | 167 (92.7) | 5 (3.5) | 0 | 5 (2.7) | 8 (5.7) | 0 | 8 (4.4) |
| | Augmentin (AUG, 30 µg) | 1 (0.7) | 0 | 1 (0.5) | 0 | 0 | 0 | 139 (99.2) | 40 (100) | 179 (99.4) |
| | Piperacillin (PIP, 30 µg) | 122 (87.1) | 34 (85) | 156 (86.6) | 1 (0.7) | 5 (12.5) | 6 (3.3) | 17 (12.1) | 1 (2.5) | 18 (10) |
| Quinolones | Nalidixic acid (NA, 30 µg) | 90 (64.2) | 5 (12.5) | 95 (52.7) | 0 | 0 | 0 | 50 (35.7) | 35 (87.5) | 85 (47.2) |
| | Noroxin (NOR, 10 µg) | 77 (55) | 1 (2.5) | 78 (43.3) | 1 (0.7) | 1 (2.5) | 2 (1.1) | 62 (44.2) | 38 (95) | 100 (55.5) |
| Aminoglycosides | Amikacin (AK, 30 µg) | 1 (0.7) | 0 | 1 (0.5) | 0 | 0 | 0 | 139 (99.2) | 40 (100) | 179 (99.4) |
| | Gentamicin (GM, 10 µg) | 8 (5.7) | 21 (52.5) | 29 (16.1) | 0 | 1 (2.5) | 1 (0.5) | 132 (94.2) | 18 (45) | 150 (83.3) |
| | Kanamycin (K, 30 µg) | 1 (0.7) | 0 | 1 (0.5) | 0 | 0 | 0 | 139 (99.2) | 40 (100) | 179 (99.4) |
| | Tobramycin (TN, 30 µg) | 17 (12.1) | 7 (17.5) | 24 (13.3) | 11 (7.8) | 3 (7.5) | 14 (7.7) | 112 (80) | 30 (75) | 142 (78.8) |
| Nitrofuran | Nitrofurantoin (FM, 50 µg) | 10 (7.1) | 3 (7.5) | 13 (7.2) | 0 | 0 | 0 | 130 (92.8) | 37 (92.5) | 167 (92.7) |
| Monobactams | Aztreonam (ATM, 30 µg) | 35 (25) | 0 | 35 (19.4) | 87 (62.1) | 21 (52.5) | 108 (60) | 18 (12.8) | 19 (47.5) | 37 (20.5) |
| Amphenicol | Chloramphenicol (C, 30 µg) | 31 (22.1) | 24 (60) | 55 (30.5) | 5 (3.5) | 0 | 5 (2.7) | 104 (74.2) | 16 (40) | 120 (66.6) |
| Sulfonamides | Trimethoprim/ sulfamethoxazole (SXT, 1.25 µg/23.75 µg) | 71 (50.7) | 19 (47.5) | 90 (50) | 0 | 0 | 0 | 69 (49.2) | 21 (52.5) | 90 (50) |
| Tetracycline | Tetracycline (30 µg) | 58 (41.4) | 30 (75) | 88 (48.8) | 0 | 0 | 0 | 82 (58.5) | 10 (25) | 92 (51.1) |

shrimp imported from China and seven (3.8%) found in frozen shrimp imported from Vietnam as shown in Table 2. The MDR (resistant to ≥3 different antimicrobials) rate was 94.4% (170/180) as shown in Table 2. The highest recorded MDR pattern to 14 different antimicrobial (AM-ATM-CIP-CAZ-CRO-CFX-GM-KF-NA-NOR-TN-PIP-SXT-TE) was detected in seven isolates which were isolated from frozen shrimp imported from China, whereas among *E. coli* isolated from frozen shrimp imported from Vietnam the highest MDR pattern to 12 different antimicrobial (AM-CIP-CTX-CRO-C-CFX-KF-NA-NOR-PIP-SXT-TE) and was found in a single isolate (Table 2). Among the 180 *E. coli* isolates were segregated into 71 different antibiotic resistance groups with MAR index ranging from 0.04 to 0.66 and 161 out of 180 (89.4%) of *E. coli* isolates recorded very significant MAR indexes above the range of 0.2 as shown in Table 2. The largest number of isolates were grouped in the12a pattern "AM-CIP-CTX-CRO-C-CFX-KF-NA-NOR-PIP-SXT-TE" with 18 (94.7%) and 1 (5.2%) isolate from the frozen shrimp imported from China and Vietnam, respectively (Table 2). Interestingly, the segregation of patterns based on the

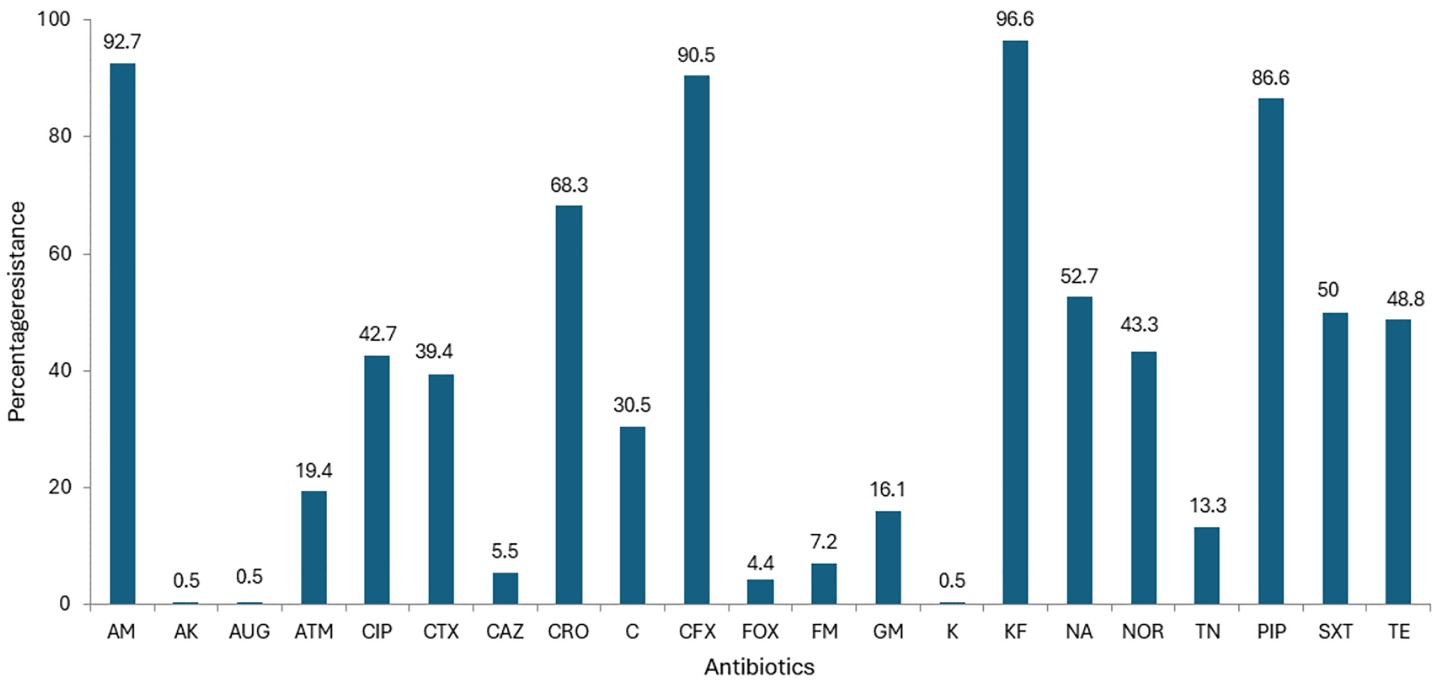

**Figure 1 Antibiotic resistance of *E. coli* isolated from imported frozen shrimp samples.**

**Table 2 Multiple antibiotic resistance and antibiotic resistance pattern exhibited by shrimp samples were presented country-wise.**

| Multiple antibiotics resistance | MAR index | Antibiotic resistance group | Antibiotic resistance pattern | China ($n = 140$) | Vietnam ($n = 40$) | No. of isolates | Total isolates |
|---|---|---|---|---|---|---|---|
| One | 0.04 | 1a | KF | 04 | 00 | 04 | 05 |
| | | 1b | CFX | 01 | 00 | 01 | |
| Two | 0.09 | 2a | AM-KF | 05 | 00 | 05 | 05 |
| Three | 0.14 | 3a | CFX-KF-PIP | 03 | 00 | 03 | 03 |
| Four | 0.19 | 4a | AM-CFX-KF-PIP | 02 | 01 | 03 | 06 |
| | | 4b | CFX-FOX-FM-KF | 02 | 00 | 02 | |
| | | 4c | AM-ATM-CRO-TN | 01 | 00 | 01 | |
| Five | 0.23 | 5a | AM-C-GM-KF-TE | 00 | 03 | 03 | 10 |
| | | 5b | AM_CRO-CFX-KF-PIP | 02 | 00 | 02 | |
| | | 5c | C-CFX-FOX-FM-KF | 01 | 00 | 01 | |
| | | 5d | AM-CFX-KF-TN-PIP | 01 | 00 | 01 | |
| | | 5e | AM-CTX-CFX-KF-PIP | 01 | 00 | 01 | |
| | | 5f | AM-C-CFX-KF-TE | 00 | 01 | 01 | |
| | | 5g | AM-CFX-GM-KF-TE | 00 | 01 | 01 | |
| Six | 0.28 | 6a | AM-CTX-CRO-CFX-KF-PIP | 08 | 06 | 14 | 23 |
| | | 6b | AM-ATM-CRO-CFX-KF-PIP | 03 | 00 | 03 | |
| | | 6c | CRO-CFX-KF-NA-PIP-SXT | 01 | 00 | 01 | |
| | | 6d | AM-CFX-KF-PIP-SXT-TE | 01 | 00 | 01 | |
| | | 6e | AM-CRO-CFX-KF-PIP-SXT | 01 | 00 | 01 | |

| Multiple antibiotics resistance | MAR index | Antibiotic resistance group | Antibiotic resistance pattern | China (n = 140) | Vietnam (n = 40) | No. of isolates | Total isolates |
|---|---|---|---|---|---|---|---|
| | | 6f | AM-CTX-CRO-CFX-FM-PIP | 00 | 01 | 01 | |
| | | 6g | AM-CIP-CFX-KF-NOR-PIP | 01 | 00 | 01 | |
| | | 6h | AM-CFX-KF-NA-NOR-PIP | 01 | 00 | 01 | |
| Seven | 0.33 | 7a | AM-CIP-CFX-K-KF-NA-NOR-PIP | 10 | 00 | 10 | 24 |
| | | 7b | AM-CRO-CFX-KF-PIP-SXT-TE | 05 | 00 | 05 | |
| | | 7c | AM-C-CFX-GM-KF-PIP-TE | 00 | 02 | 02 | |
| | | 7d | AM-CRO-CFX-KF-NA-PIP-SXT | 02 | 00 | 02 | |
| | | 7e | CIP-CRO-CFX-KF-NA-NOR-PIP | 01 | 00 | 01 | |
| | | 7f | AM-CTX-CRO-CFX-K-KF-PIP | 01 | 00 | 01 | |
| | | 7g | AM-CTX-CRO-CFX-FM-PIP-SXT | 00 | 01 | 01 | |
| | | 7h | AM-CTX-CRO-C-CFX-KF-PIP | 00 | 01 | 01 | |
| | | 7i | AM-ATM-CTX-CRO-CFX-KF-PIP | 01 | 00 | 01 | |
| Eight | 0.38 | 8a | AM-CIP-CRO-CFX-KF-NA-NOR-PIP | 12 | 00 | 12 | 40 |
| | | 8b | AM-CTX-CRO-CFX-KF-PIP-SXT-TE | 03 | 05 | 08 | |
| | | 8c | AM-ATM-CRO-CFX-KF-NA-PIP-SXT | 08 | 00 | 08 | |
| | | 8d | AM-CIP-KF-NA-NOR-TN-PIP-TE | 04 | 00 | 04 | |
| | | 8e | AM-CTX-C-CFX-GM-KF-PIP-TE | 00 | 01 | 01 | |
| | | 8f | AM-AK-CIP-CFX-KF-NA-NOR-PIP | 01 | 00 | 01 | |
| | | 8g | AM-CIP-CAZ-CFX-KF-NA-NOR-PIP | 01 | 00 | 01 | |
| | | 8h | AM-CTX-CRO-CFX-TN-PIP-SXT-TE | 00 | 01 | 01 | |
| | | 8i | AM-CIP-CRO-CFX-KF-NA-NOR-SXT | 01 | 00 | 01 | |
| | | 8j | AM-ATM-CRO-CFX-KF-PIP-SXT-TE | 01 | 00 | 01 | |
| | | 8k | AM-CRO-CFX-FM-KF-NA-PIP-SXT | 01 | 00 | 01 | |
| | | 8m | AM-ATM-CTX-CRO-CFX-PIP-SXT-TE | 01 | 00 | 01 | |
| Nine | 0.42 | 9a | AM-CTX-CRO-C-CFX-GM-KF-PIP-TE | 00 | 03 | 03 | 11 |
| | | 9b | AM-C-CFX-GM-KF-NA-PIP-SXT-TE | 00 | 02 | 02 | |
| | | 9c | AM-ATM-CIP-CRO-CFX-KF-NA-NOR-PIP | 02 | 00 | 02 | |
| | | 9d | AM-CTX-CRO-C-CFX-GM-KF-TN-TE | 00 | 01 | 01 | |
| | | 9e | AM-C-CFX-GM-KF-TN-PIP-SXT-TE | 00 | 01 | 01 | |
| | | 9f | AM-ATM-C-CFX-FOX-FM-KF-NA-SXT | 01 | 00 | 01 | |
| | | 9g | AM-ATM-CAZ-CRO-CFX-KF-NA-PIP-SXT | 01 | 00 | 01 | |
| Ten | 0.47 | 10a | AM-CTX-CRO-C-CFX-GM-KF-PIP-SXT-TE | 00 | 02 | 02 | 08 |
| | | 10b | AM-CRO-C-CFX-GM-KF-NA-PIP-SXT-TE | 00 | 01 | 01 | |
| | | 10c | AM-CTX-CRO-C-CFX-FM-KF-PIP-SXT-TE | 00 | 01 | 01 | |
| | | 10d | AM-CTX-C-CFX-GM-KF-TN-PIP-SXT-TE | 00 | 01 | 01 | |
| | | 10e | AM-C-CFX-GM-KF-NA-TN-PIP-SXT-TE | 00 | 01 | 01 | |
| | | 10f | AM-ATM-CIP-CFX-KF-NA-NOR-PIP-SXT-TE | 01 | 00 | 01 | |
| | | 10g | AM-AUG-ATM-C-CFX-FOX-FM-KF-NA-SXT | 01 | 00 | 01 | |

| Table 2 (continued) | | | | | | | |
|---|---|---|---|---|---|---|---|
| Multiple antibiotics resistance | MAR index | Antibiotic resistance group | Antibiotic resistance pattern | China (n = 140) | Vietnam (n = 40) | No. of isolates | Total isolates |
| Eleven | 0.52 | 11a | AM-CIP-CRO-C-CFX-KF-NA-NOR-PIP-SXT-TE | 03 | 00 | 03 | 08 |
| | | 11b | AM-CTX-CRO-C-CFX-GM-KF-TN-PIP-SXT-TE | 00 | 02 | 02 | |
| | | 11c | AM-CIP-CTX-CRO-CFX-KF-NA-NOR-PIP-SXT-TE | 02 | 00 | 02 | |
| | | 11d | AM-ATM-CIP-CFX-FOX-FM-KF-NOR-PIP-SXT-TE | 01 | 00 | 01 | |
| Twelve | 0.57 | 12a | AM-CIP-CTX-CRO-C-CFX-KF-NA-NOR-PIP-SXT-TE | 18 | 01 | 19 | 20 |
| | | 12c | AM-ATM-CIP-CRO-CFX-FOX-FM-KF-NA-NOR-SXT-TE | 01 | 00 | 01 | |
| Thirteen | 0.61 | 13a | AM-ATM-CIP-CTX-CRO-C-CFX-KF-NA-NOR-PIP-SXT-TE | 02 | 00 | 02 | 10 |
| | | 13b | AM-ATM-CIP-CTX-CRO-CFX-KF-NA-NOR-TN-PIP-SXT-TE | 02 | 00 | 02 | |
| | | 13c | AM-CIP-CTX-CRO-C-CFX–FM-KF-NA-NOR-PIP-SXT-TE | 02 | 00 | 02 | |
| | | 13d | AM-CIP-CTX-CRO-C-CFX-KF-NA-NOR-TN-PIP-SXT-TE | 01 | 00 | 01 | |
| | | 13e | AM-ATM-CIP-CAZ-CRO-CFX-KF-NA-NOR-TN-PIP-SXT-TE | 01 | 00 | 01 | |
| | | 13f | AM-CIP-CTX-CRO-C-CFX-FOX-KF-NA-NOR-PIP-SXT-TE | 01 | 00 | 01 | |
| | | 13g | AM-CIP-CTX-CRO-C-CFX-GM-KF-NA-NOR-PIP-SXT-TE | 01 | 00 | 01 | |
| Fourteen | 0.66 | 14a | AM-ATM-CIP-CAZ-CRO-CFX-GM-KF-NA-NOR-TN-PIP-SXT-TE | 07 | 00 | 07 | 07 |
| Total | | 71 | | 140 | 40 | 180 | 180 |

source of country origin of frozen shrimp revealed few inter-country resistant patterns found interconnecting such as 4a, 6a, 8b, and 12a patterns, which were analogous in both the countries. This influenced 44 (24.4%) isolates overlapping between China and Vietnam as shown in Fig. 2 and Table 3. The depicted results are more exciting as all four overlapping patterns belong to different numbers of MAR groups (Table 3).

## Antimicrobial co-resistance patterns

Upon evaluation of co-resistance results, it was found that the highest co-resistance of 162 (90%) isolates was observed between cephalothin and ampicillin, followed by 158 (87.7%) isolates between cephalothin and Cephalexin as shown in Table 4. The low co-resistance was found between amikacin and all other tested antibiotics showed 0.5%, and similar results were obtained for augmentin as shown in Table 4.
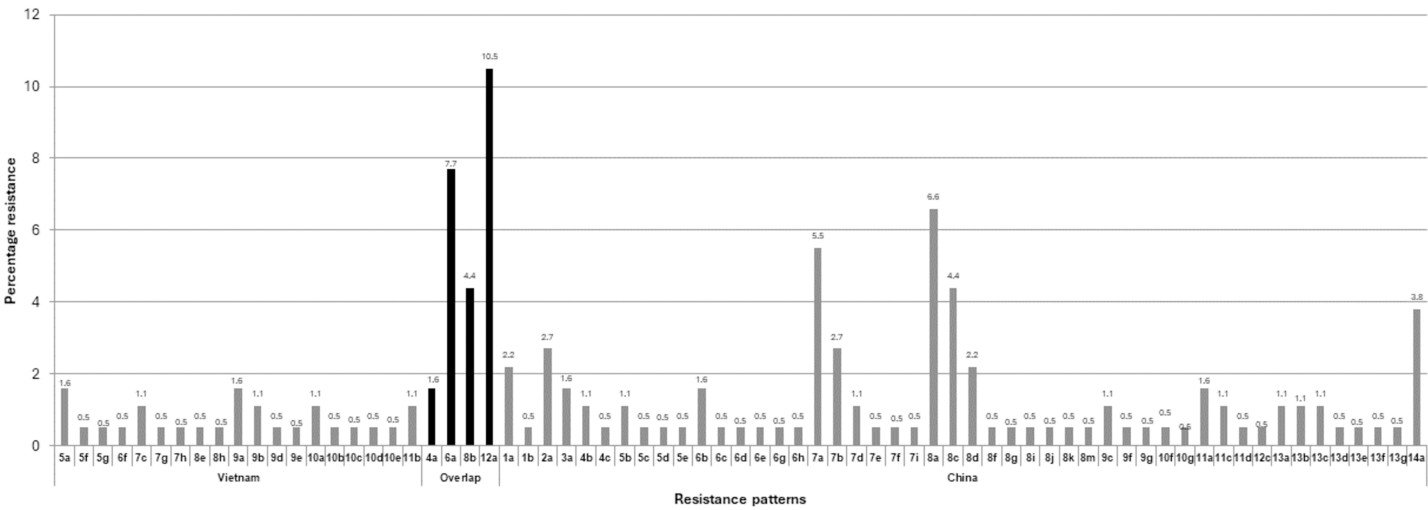

**Figure 2 Country-wise segregation of antibiotic resistance patterns with inter-country analogous patterns.**

**Table 3 Segregation of patterns by country-wise and presenting inter-country interconnecting resistance patterns.**

| Country | Resistance pattern | No. of patterns | No. of isolates |
|---|---|---|---|
| Vietnam | 5a, 5f, 5g, 6f, 7c, 7g, 7h, 8e, 8h, 9a, 9b, 9d, 9e, 10a, 10b, 10c, 10d, 10e, 11b, | 19 | 27 |
| Overlap | 4a, 6a, 8b, 12a | 04 | 44 |
| China | 1a, 1b, 2a, 3a, 4b, 4c, 5b, 5c, 5d, 5e, 6b, 6c, 6d, 6e, 6g, 6h, 7a, 7b, 7d, 7e, 7f, 7i, 8a, 8c, 8d, 8f, 8g, 8i, 8j, 8k, 8m, 9c, 9f, 9g, 10f, 10g, 11a, 11c, 11d, 12c, 13a, 13b, 13c, 13d, 13e, 13f, 13g, 14a | 48 | 109 |
| Total | | 71 | 180 |

# DISCUSSION

Seafood products serve as the primary food source because of their high polyunsaturated fatty acids and essential trace elements (*Mesa et al., 2021*). Consequently, numerous coastal nations have promoted seafood farming to fulfill both local and international demand (*Alshubiri, Elheddad & Doytch, 2020*). The contamination of seafood with antibiotics and antibiotic-resistant bacteria poses a potential threat to human health (*Hossain et al., 2022*). A recent investigation revealed that European countries confiscated seafood and seafood products from four Southeast Asian countries (Malaysia, Thailand, Vietnam, and Indonesia) due to the presence of antibiotics, and these products were rejected by 19 European countries from 1997 to 2020 due to the presence of pathogens and antibiotics (*Odeyemi et al., 2023a*, *2023b*). Seafood-borne pathogens and antibiotic resistance are recognized as global health concerns since they jeopardize food security (*Himanshu et al., 2022*). In Europe, antibiotic-resistant pathogens have been linked to over 30,000 deaths annually (*Himanshu et al., 2022*). It has been projected that by 2050, antibiotic-resistant pathogens could lead to 10 million deaths per year globally causing, potentially, a reduction of 2% to 5% in the gross domestic product, which amounts to approximately 100 trillion US dollars (*de Kraker, Stewardson & Harbarth, 2016*).

**Table 4** Co-resistance patterns exhibited by *E. coli* isolates from shrimp samples.

| Antibiotic | AM | AK | AUG | ATM | CIP | CTX | CAZ | CRO | C | CFX | FOX | FM | GM | K | KF | NA | NOR | TN | PIP | SXT |
|---|---|---|---|---|---|---|---|---|---|---|---|---|---|---|---|---|---|---|---|---|
| AK | 1 (0.5) | | | | | | | | | | | | | | | | | | | |
| AUG | 1 (0.5) | 0 | | | | | | | | | | | | | | | | | | |
| ATM | 35 (19.4) | 0 | 1 (0.5) | | | | | | | | | | | | | | | | | |
| CIP | 76 (42.2) | 1 (0.5) | 0 | 17 (9.4) | | | | | | | | | | | | | | | | |
| CTX | 71 (39.4) | 0 | 0 | 6 (3.3) | 30 (16.6) | | | | | | | | | | | | | | | |
| CAZ | 10 (5.5) | 0 | 0 | 9 (5.0) | 9 (5.0) | 0 | | | | | | | | | | | | | | |
| CRO | 121 (67.2) | 0 | 0 | 31 (17.2) | 58 (32.2) | 68 (37.7) | 9 (5.0) | | | | | | | | | | | | | |
| C | 54 (30.0) | 0 | 1 (0.5) | 4 (2.2) | 29 (16.1) | 38 (21.1) | 0 | 40 (22.2) | | | | | | | | | | | | |
| CFX | 154 (85.5) | 1 (0.5) | 1 (0.5) | 34 (18.8) | 73 (40.5) | 71 (39.4) | 10 (5.5) | 122 (67.7) | 52 (28.8) | | | | | | | | | | | |
| FOX | 5 (2.7) | 0 | 1 (0.5) | 4 (2.2) | 3 (1.6) | 1 (0.5) | 0 | 2 (1.1) | 4 (2.2) | 8 (4.4) | | | | | | | | | | |
| FM | 10 (5.5) | 0 | 1 (0.5) | 4 (2.2) | 4 (2.2) | 5 (2.7) | 0 | 7 (3.8) | 6 (3.3) | 13 (7.2) | 7 (3.8) | | | | | | | | | |
| GM | 29 (16.1) | 0 | 0 | 7 (3.8) | 8 (4.4) | 11 (6.1) | 7 (3.8) | 17 (9.4) | 21 (11.6) | 26 (14.4) | 0 | 0 | | | | | | | | |
| K | 1 (0.5) | 0 | 0 | 0 | 0 | 1 (0.5) | 0 | 1 (0.5) | 0 | 1 (0.5) | 0 | 0 | 0 | | | | | | | |
| KF | 162 (90.0) | 1 (0.5) | 1 (0.5) | 33 (18.3) | 77 (42.7) | 67 (37.2) | 10 (5.5) | 118 (65.5) | 55 (30.5) | 158 (87.7) | 8 (4.4) | 11 (6.1) | 29 (16.1) | 1 (0.5) | | | | | | |
| NA | 93 (51.6) | 1 (0.5) | 1 (0.5) | 27 (15.0) | 75 (41.6) | 30 (16.6) | 10 (5.5) | 72 (40.0) | 35 (19.4) | 91 (50.5) | 4 (2.2) | 6 (3.3) | 12 (6.6) | 0 | 95 (52.7) | | | | | |
| NOR | 77 (42.7) | 1 (0.5) | 0 | 17 (9.4) | 77 (42.7) | 30 (16.6) | 9 (5.0) | 58 (32.2) | 29 (16.1) | 74 (41.1) | 3 (1.6) | 4 (2.2) | 8 (4.4) | 0 | 78 (43.3) | 76 (42.2) | | | | |
| TN | 24 (13.3) | 0 | 0 | 11 (6.1) | 15 (8.3) | 8 (4.4) | 8 (4.4) | 16 (8.8) | 7 (3.8) | 19 (10.5) | 0 | 0 | 13 (7.2) | 0 | 22 (12.2) | 16 (8.8) | 15 (8.3) | | | |
| PIP | 151 (83.8) | 1 (0.5) | 0 | 31 (17.2) | 75 (41.6) | 70 (38.8) | 10 (5.5) | 119 (66.1) | 47 (26.1) | 152 (84.4) | 2 (1.1) | 7 (3.8) | 24 (13.3) | 1 (0.5) | 152 (84.4) | 91 (50.5) | 76 (42.2) | 22 (12.2) | | |
| SXT | 89 (49.4) | 0 | 1 (0.5) | 28 (15.5) | 45 (25.0) | 47 (26.1) | 9 (5.0) | 80 (44.4) | 42 (23.3) | 90 (50.0) | 5 (2.7) | 9 (5.0) | 18 (10.0) | 0 | 87 (48.3) | 63 (35.0) | 45 (25.0) | 17 (9.4) | 86 (47.7) | |
| TE | 88 (48.8) | 0 | 0 | 17 (9.4) | 48 (26.6) | 51 (28.3) | 8 (4.4) | 68 (37.7) | 51 (28.3) | 81 (45.0) | 3 (1.6) | 5 (2.7) | 29 (16.1) | 0 | 86 (47.7) | 51 (28.3) | 48 (26.6) | 22 (12.2) | 81 (45.0) | 72 (40.0) |

A recently published review article from Singapore highlighted antimicrobial resistance among bacteria originating from 11 Southeast Asian countries. It revealed that most antimicrobial resistance reports have come from Vietnam, Malaysia, and Thailand, respectively (*Suyamud et al., 2023*). Also, the antimicrobial resistance found in Southeast Asian aquaculture was classified into 17 drug classes (*Suyamud et al., 2023*). The most reported antimicrobial resistance are aminoglycosides, beta-lactams, (fluoro) quinolones, tetracycline, sulpha group, and multi-drug resistance (*Suyamud et al., 2023*). The same study revealed that beta-lactams, tetracycline, and sulpha groups are reported in each country with frequencies higher than 40% and the most widely and frequently reported antimicrobial resistance were found in Southeast Asian aquaculture are strains of *E. coli*, *Aeromonas*, and *Vibrio* (*Suyamud et al., 2023*). Notably, *E. coli* isolates in our study exhibited high resistance against the antibiotic classes of cephalosporins, penicillins, quinolones, sulfonamides, and tetracycline. Our findings are consistent with earlier research study from China conducted on AMR prevalence of MDR *E. coli* in retail aquatic products available for purchase such as shrimp, fish and shellfish, where the study revealed that 40% of overall investigated samples were contaminated with MDR *E. coli* (*Zhang et al., 2024*). Additionally, the study reported that the *E. coli* isolates showed high prevalence of resistance to tetracycline (93.7%), trimethoprim-sulfamethoxazole (78.9%), ampicillin (78.4%), chloramphenicol (72.1%), nalidixic acid (73.2%), cephalothin (65.3%), streptomycin (65.8%), kanamycin (42.1%), gentamicin (37.9%), ciprofloxacin (42.6%), and norfloxacin (45.8%) (*Zhang et al., 2024*). Whereas, in our study isolates of *E. coli* isolated from frozen shrimp imported from China showed high resistance to cephalotin (97.8%), ampicillin (90.7%), cephalexin (90%), piperacillin (87.1%), ceftriaxone (69.2%), nalidixic acid (64.2%), noroxin (55%), trimethoprim/sulfamethoxazole (50.7%), and tetracycline (41.4%).

Our study is in concordance with recent research from the USA which reported high antimicrobial resistance among *E. coli* isolated from imported shrimp. The study revealed that the isolates showed resistance to eight different antibiotic classes namely gentamicin, streptomycin, ampicillin, chloramphenicol, nalidixic acid, ciprofloxacin, tetracycline, and trimethoprim/sulfamethoxazole (*Sung et al., 2024*). Furthermore, another similar study from the USA highlighted quinolone resistance *E. coli* isolated from imported shrimp as with 52.3% of isolates exhibited resistance against nalidixic acid, ampicillin, tetracycline and chloramphenicol (*Nawaz et al., 2015*). Another study from Vietnam investigated antimicrobial resistance in total of 88 *E. coli* strains that were isolated from wild and farm fish and results revealed that a high prevalence of 94.3% of isolates were resistant to sulfonamides (*Hoa et al., 2020*). The usage of different classes of antibiotics, such as β-lactams, quinolones, sulfonamides, tetracyclines, and nitrofuran are among the most commonly used antibiotics in aquaculture and shrimp farming worldwide (*Pepi & Focardi, 2021*; *Sharma et al., 2021*). However, the consequences of the use of antibiotics in aquaculture for both therapeutic and prophylaxis may enhance the bacteria found within the water environment to develop antibiotic resistance and these resistance genes further spread through horizontal gene transfer to surrounding bacteria (*Pepi & Focardi, 2021*). Therefore, antibiotic usage in the aquaculture industry is expected to rise since aquaculture

industry is rapidly growing worldwide and particularly in Southeast Asia to provide safe food for the growing global population that is expected to increase to about 10 billion people by 2030 (*Santos & Ramos, 2018*).

The rise and emergence of antimicrobial resistance is a globally acknowledged issue according to the Food and Agriculture Organization of the United Nations (FAO) Action Plan on Antimicrobial Resistance (AMR) 2016–2020. The FAO Action Plan aims to assist the agricultural and food sectors in addressing AMR on an international scale (*Food and Agriculture Organization of the United Nations, 2016*). The FAO has identified that the threat of increasing AMR is more pronounced in nations with inadequate legislation and regulatory frameworks governing the use of antimicrobial agents, compared to those with established action plans and monitoring systems for antimicrobial usage (*Thornber et al., 2020*). Furthermore, the global food trade is likely contributed to the spread of AMR across borders, with well-regulated countries potentially facing the risk of introducing new resistant bacterial pathogens harboring resistance genes in plasmids and transposons, thereby increasing the national AMR burden through imported food products (*Ellis-Iversen et al., 2020*). Due to the high levels of international trade and the direct links to aquatic ecosystems, shrimp aquaculture may facilitate the global spread of AMR. Most shrimp production takes place in developing countries, where antibiotic quality and usage are often poorly regulated. Additionally, in shrimp farming regions, untreated waste is frequently discharged directly into local water bodies (*Thornber et al., 2020*). These risks contrast sharply with those associated with other major aquaculture products, such as salmon, which are cultivated in higher-income countries with stricter regulations and established management practices. In contrast, several early studies investigated AMR in shrimp concluded that evaluating the true extent of AMR risk in the shrimp sector is a significant challenge, particularly due to the difficulty in obtaining accurate data on antibiotic use (*Sivaraman et al., 2021*; *Loest et al., 2022*; *Sung et al., 2024*; *Thaotumpitak et al., 2024*). Consequently, a recent research article reviewed critically the potential risks of antimicrobial resistance in the global shrimp industry revealing that assessing the risks associated with antimicrobial use in this rapidly expanding sector is currently quite challenging, because it includes diverse production systems that exist at the intersection of aquatic and terrestrial environments (*Thornber et al., 2020*). In addition, the study addressed the risks linked to AMR is further complicated by the trend toward intensification and the accompanying disease pressures, as many farmers currently lack alternatives to antibiotics for preventing crop losses (*Thornber et al., 2020*).

In this study, *E. coli* isolates displayed a ranging of multiple antibiotic resistance (MAR) index values ranged from one antimicrobial resistant to 14 antimicrobials. Studies demonstrated that bacteria with a MAR index exceeding 0.2 are typically linked to high-risk sources of contamination indicating, also, high usage of antibiotic growth promoters. Thus, in this study 89.4% of *E. coli* isolates exhibited MAR index in the range of 0.2 to 0.66 indicating that the majority of isolates originated from sources of high antibiotic exposures. A significantly larger proportion of *E. coli* isolates displayed MDR with an overall rate of 94.4%. The overall resistance to antimicrobials in this study ranged between resistance to one antimicrobial and 14 different classes of antimicrobials. The highest

number of MDR was found among 8, 18 and 7 *E. coli* isolates that were resistant to 8, 12 and 14 different classes of antimicrobials, respectively, in frozen shrimp imported from China. In comparison, 6, 5, 3, 2 and one *E. coli* isolates exhibited resistance to 6, 8, 9, 11, and 12 resistant different classes of antimicrobials, respectively, in frozen shrimp imported from Vietnam. Isolation of MDR *E. coli* from shrimp, seafood and other seafood products has been reported by several studies in different countries (*Loest et al., 2022*; *Caputo et al., 2023*; *Odeyemi et al., 2023a*). A study from India reported high prevalence of multiple antibiotic-resistant *E. coli* isolated from fresh seafood sold in retail markets of Mumbai (*Singh, Nayak & Kumar, 2020*). However, the same study revealed that more than 90% of isolates were resistant to cephalosporins (cefotaxime, cefpodoxime, and ceftazidime) and MAR index of 97.35% of the isolates was above 0.18 (*Singh, Nayak & Kumar, 2020*).

The fast-expanding aquaculture industry depends significantly on antimicrobials to prevent infectious bacterial diseases that pose risks to production, thereby researchers have been investigating antimicrobial resistance in aquaculture for over five decades and notably reported rise in evidence concerning antimicrobial resistance within this sector (*Heuer et al., 2009*; *Reverter et al., 2020*; *Caputo et al., 2023*). In Saudi Arabia, very limited studies investigated the antimicrobial resistance in imported frozen aquaculture fishery products. Early studies from Saudi Arabia investigated antimicrobial resistance in retail imported frozen freshwater fish studies revealed high rate of antimicrobial resistance among isolates of *Salmonella* spp. and *E. coli* isolated (*Elhadi, 2014*, *2016*). Antimicrobial resistance (AMR) in aquaculture has the potential to be transmitted to clinically significant strains in the natural environment *via* horizontal gene transfer, which can have consequences for the entire ecosystem (*Preena et al., 2020*). Moreover, many studies have confirmed that shrimp aquaculture harbor bacterial pathogens that demonstrate multiple antibiotic resistance (*Preena et al., 2020*). Imported frozen shrimp and other aquaculture products can serve as potential carriers for the spread of clinically significant antimicrobial-resistant bacteria and genes associated with resistance, such as extended-spectrum ß-lactamases (ESBLs), plasmid-mediated quinolone resistance determinants (PMQR), colistin resistance (mcr-1), and carbapenemases (*Jung, Morrison & Rubin, 2022*; *Loest et al., 2022*).

## CONCLUSION

The presence of multi-drug resistance *E. coli* in imported frozen shrimp available for purchase in the Eastern Province of Saudi Arabia indicates possible of unsanitary practices and this contamination may pose a risk of human infection. The obtained results in this study indicate the emergence of resistance and a decline in the efficacy of antimicrobial agents. Additionally, 94.4% of examined isolates exhibited MDR and 90% of isolates showed co-resistance between cephalotin and ampicillin, followed by 87.7% co-resistance between cephalotin and cephalexin. Moreover, the obtained MAR index values in this study revealed that isolates of *E. coli* isolated from imported frozen shrimp originated from high sources of antibiotics exposures and were used in large amounts or to a great degree. Such findings underline the need for collaborative efforts between scientists and food authorities in Saudi Arabia to work together to monitor presence of antimicrobial

resistance bacteria in imported frozen shrimp and other aquaculture products. Implementing hygienic practices among imported frozen aquaculture products is recommended to decrease the transmission of antimicrobial-resistant *E. coli* and other bacterial species within the human food chain. This study addresses the need for further research in Saudi Arabia to aid in monitoring and investigation of AMR bacteria in imported frozen aquaculture products. However, our study has two limitations: firstly, the number of samples examined were not optimal, and secondly, screening for antibiotic-resistance genes was not performed.

### Funding

This work was supported by the Deanship of Scientific Research, Imam Abdulrahman Bin Faisal University (Grant No. 2014191). The funders had no role in study design, data collection and analysis, decision to publish, or preparation of the manuscript.

### Grant Disclosures

The following grant information was disclosed by the authors:
Deanship of Scientific Research, Imam Abdulrahman Bin Faisal University:  2014191.

### Competing Interests

The authors declare that they have no competing interests.

### Author Contributions

- Ibrahim Alhabib performed the experiments, analyzed the data, prepared figures and/or tables, authored or reviewed drafts of the article, and approved the final draft.
- Nasreldin Elhadi conceived and designed the experiments, performed the experiments, analyzed the data, prepared figures and/or tables, authored or reviewed drafts of the article, and approved the final draft.

### Data Availability

    The raw data is available in the Supplemental File.

### Supplemental Information

Supplemental information for this article can be found online at http://dx.doi.org/10.7717/peerj.18689#supplemental-information.

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
