# Peer review of "Antimicrobial resistance pattern of Escherichia coli isolated from imported frozen shrimp in Saudi Arabia"

_PeerJ, doi:10.7717/peerj.18689_

## Round 0.1 · original submission · Major Revisions

Please revise your manuscript based on the reviewers' comments. When re-submitting your revised manuscript, please make sure a point-by-point response letter is also attached, showing that all the questions and comments have been adequately addressed.

Reviewer 1 ·

Basic reporting

-The paper language is clear

-The quality of the figures and tables is outstanding and easy to appreciate

Experimental design

Would you please explain why you don't perform antimicrobial resistance gene detection, it will make your research more relevant ??

Validity of the findings

no comment

Additional comments

Please find attached some mistakes to correct
Abstract
Line 30 : E.coli in italic
Line 35 : E.coli in italic
Line 38 : E.coli in italic
Introduction
Line 73 : E.coli in italic
Line 98 : (BIOHAZ) et al., 2021: please check the reference style.
Line 112 : national action plans : Capitalise word
Materiel & methods
Line 172 : E.coli in italic
Line 182 : Antibiotic : lowercase letter
Results
Line 216 : E.coli in italic
Discussion
Line 266 : E.coli in italic
Line 271 : E.coli in italic (2 words in the same line)

Annotated reviews are not available for download in order to protect the identity of reviewers who chose to remain anonymous.

Reviewer 2 ·

Basic reporting

Local Context in the Introduction: It would be helpful to include some data or statistics on antimicrobial resistance in Saudi Arabia to provide more local context in the introduction.

Visualization of Results: I suggest using a heatmap to visually represent the data currently in Table 2. This could make it easier for readers to interpret patterns of antimicrobial resistance across different isolates.

Typographical Errors:
* Line 308: “Dute” should be corrected to “Due.”
* In the abstract, ensure the formatting is consistent for the reported values. For example, “cephalothin 174 (96.6%), ampicillin 167 (92.7%)” should be written as “cephalothin (174/180, 96.6%), ampicillin (167/180, 92.7%).”

Experimental design

Overall, the manuscript is okay. My main concern is the sample size and the absence of genetic testing methods (such as PCR or whole genome sequencing) that could provide a deeper understanding of the AMR genes and mutations.

Details on Shrimp Origins: While the manuscript mentions that shrimp samples were purchased from different supermarkets in Al Khobar, it would be beneficial to clarify which regions of China and Vietnam the samples originated from. Were they from different brands or producers? This information could add valuable depth to the analysis.

Validity of the findings

Expansion of Antimicrobial Co-Resistance Analysis: The interpretation and discussion of the antimicrobial co-resistance results could be expanded. Is it possible to identify additional antibiotics that showed noteworthy co-resistance? Additionally, were there any significant differences in co-resistance patterns between the samples from China and Vietnam? Discussion — Compare findings with other studies

Comparison with previous studies: The authors conducted similar studies in 2014 and 2016 (Elhadi, 2014, 2016). It would be useful to compare the current results with those previous studies to discuss any trends, changes, or similarities in AMR patterns over time.

Mention Limitations in the Discussion: It would be important to acknowledge the limitations of the study, particularly the sample size and the lack of genetic testing (e.g., PCR or whole genome sequencing), in the discussion section.

Saudi Food and Drug Authority (SFDA) Regulations: While the manuscript mentions that the SFDA requires health certificates for imported seafood, the findings suggest that the current health certificate requirements may not be comprehensive. It would be useful to discuss this observation and offer some suggestions on how the regulatory framework could be improved.

Additional comments

The manuscript presents an important study on the antimicrobial resistance patterns of Escherichia coli isolated from imported frozen shrimp in Saudi Arabia. This research addresses a critical public health concern, particularly given the rising rates of antimicrobial resistance (AMR) and the potential implications for food safety and human health.

Overall, the manuscript is okay. My main concern is the sample size and the absence of genetic testing methods (such as PCR or whole genome sequencing) that could provide a deeper understanding of the AMR genes and mutations.

Below are some additional suggestions and areas for improvement:
1. Local Context in the Introduction: It would be helpful to include some data or statistics on antimicrobial resistance in Saudi Arabia to provide more local context in the introduction.
2. Details on Shrimp Origins: While the manuscript mentions that shrimp samples were purchased from different supermarkets in Al Khobar, it would be beneficial to clarify which regions of China and Vietnam the samples originated from. Were they from different brands or producers? This information could add valuable depth to the analysis.
3. Visualization of Results: I suggest using a heatmap to visually represent the data currently in Table 2. This could make it easier for readers to interpret patterns of antimicrobial resistance across different isolates.
4. Expansion of Antimicrobial Co-Resistance Analysis: The interpretation and discussion of the antimicrobial co-resistance results could be expanded. Is it possible to identify additional antibiotics that showed noteworthy co-resistance? Additionally, were there any significant differences in co-resistance patterns between the samples from China and Vietnam? Discussion — Compare findings with other studies
5. Comparison with previous studies: The authors conducted similar studies in 2014 and 2016 (Elhadi, 2014, 2016). It would be useful to compare the current results with those previous studies to discuss any trends, changes, or similarities in AMR patterns over time.
6. Mention Limitations in the Discussion: It would be important to acknowledge the limitations of the study, particularly the sample size and the lack of genetic testing (e.g., PCR or whole genome sequencing), in the discussion section.
7. Saudi Food and Drug Authority (SFDA) Regulations: While the manuscript mentions that the SFDA requires health certificates for imported seafood, the findings suggest that the current health certificate requirements may not be comprehensive. It would be useful to discuss this observation and offer some suggestions on how the regulatory framework could be improved.
8. Typographical Errors:
* Line 308: “Dute” should be corrected to “Due.”
* In the abstract, ensure the formatting is consistent for the reported values. For example, “cephalothin 174 (96.6%), ampicillin 167 (92.7%)” should be written as “cephalothin (174/180, 96.6%), ampicillin (167/180, 92.7%).”

·

Basic reporting

The written English is not very clear in some areas and some incorrect text were used, which made the narration unclear

Experimental design

The research question is well defined, relevant, & meaningful.
The rigorousness of the investigation performed is not technically high, as molecular confirmation of E. coli was not carried out.

The antibiotic susceptibility test method described does not provide sufficient detail & information to replicate it.

Validity of the findings

The findings are very valid.

Additional comments

Abstract section:
Line 4 , “E. coli” should be written in full. The first word of a sentence should not be abbreviated. They should be written in full.
Line 7: “….34 tested positive” This does not tally with the result section where it was said that “30” samples tested positive for E.coli. Confirm and reconcile accordingly.
Line 9 – 10: “ ….23 out of 25 and 12 out of 15” i.e 23 +12 = 35 not 34 as stated in line 6. Reconcile the figures accordingly.

Line 12, 17, 20: “E.coli” should be italicized all through the manuscript draft
Line 17: “ resistant to ≥3 different antimicrobials” should be written as a different “class” of antimicrobials.

General comments: The abstract section is too long ~500 words. The abstract should be reduced and only important data should be included in the abstract section.

Introduction section

Line 45: “ …in nature” should be removed
Line 46: “…despite the fact that” should be changed to “even though”
Line 51: “E. coli” should be written in full. The first word of a sentence should not be abbreviated. They should be written in full.
Line 54: what are you referring to as “ This” ?
Line 54-56: recast this sentence for more clarity
Line 62: humans to be changed to” human”
Line 66: …..common to be changed to “..commonly”
Line 67: …..Pitout et al., 1998 reference is too old
Line 79: “..the spread”
Line 87: “….. depend” to be changed to “dependent”
Line 103: “…. in an efficient way” change to “… efficiently”
Line 107: “….. as assessing” change to “assess”


MATERIALS AND METHODS SECTION

Line 126: E. coli” should be written in full. The first word of a sentence should not be abbreviated. E.coli should be italicized.
Line 127: “ was” change to were
Line 128: “…samples” change to sample
Line 132: “ …were incubated”

Antibiotic susceptibility Testing
Line 140 -141What was the standard concentration used for the turbidity of the suspension? How are you sure that the turbidity was accurate for the assay?
Line 154: “…. reference strains will be used as control in perform” recast this statement. The research has already been done and not at the proposal stage.
Line 154: E.coli should be italicized
Line 157: “…an isolates” should be changed to either “isolates” or “an isolate”
Line 165-168: there is no consistency in the reported result compare with the abstract. Check the number of positive samples “30” reported here compared with the 34/35in the abstract section.
Line 175: check the spelling for “ ..cephalotin” cephalothin
Line 177: “… trimethoprim-Sulphamethoxazole”
Line 178: remove the from “…the higher”
Line 185: “..drug resistance”
Line 188: “…was found”
Line 190: “… antimicrobials”
Line 198: “ ….ranging”
Line 199. “ E.coli”
Line 205-207: the sentences are not clear. Why the word “exciting” ? this is not appropriate in this context
Line 213 -215: this statement here is not clear. What exactl are your trying to say? Is the 100% susceptibility a “co-resistance”

Discussion section
Line 225-239 should be removed. The section should go straight to the discussion of the result and not review of literature. This section can be merged/harmonized with the introduction section.
Your discussion should start with just an introductory sentence after which you go straight to the discussion of your result.
Line 241: “… originating”
Line 249, 254: “…E.coli” italicized
Line 250: “…findings are”
Line 251: “..on AMR”
Line 261: “…Noroxin”
Line 262: “ ….USA which reported….” Is the result stated here referring to the USA study? The statement is not clear, recast.
Line 266: “Furthermore
Line 266 – 269: what do you mean by “..early research from USA…” this statement is not clear
Line 278: “…is rapidly
Line 279: “… Southeast Asia”
Line 283: “ The FAO..”
Line 291 : “Due”
Line 310: “…ranging”
Line 315: “.. in the range”
Line 318: remove “were”
Line 324: “… has been”
Line 336: “…in retail
Line 337: remove “ isolated”
Line 344: extended-spectrum
Line 350: “…the Eastern” remove “of”
Line 351: “….pose a risk” “…..the emergence”
Line 352: “….resistance..”
Line 353: showed
Line 356: and were

·

Basic reporting

1. Its clear and unambiquous but required grammar improvement
2. Lierature references are ok and context provided
3.Professional article structure, figures, and tables are fine.
Self contained with results.
4. Hypotheses not involved

Experimental design

1. Aims and scope is for the ARB and ARGs
2. Research questions are defined, and relevant
3. No ethics involved
4. Methods are ok. Explains in detail about the results of API 20E for the confirmation of E. coli pathogenic strains. Why did you select 180 isolates from 40 samples and did AST?
Delete line no 157- 158.

What is reason for carrying out AST in all 180 number of E.coli strains? is they varied in biovars? Is the each and every isolate varied in AST profiles?
Delete line 204-206 and is not required.

Validity of the findings

1. Did good works on the surveillance ARB and its AST from the imported shrimp sample and is need of the hours since AMR is an emerging global health issues.
Line no.185-186 report the incidences (%) of E. coli from the China & Vietnam samples.
What is reason for carrying out AST in all 180 number of E.coli strains? is they varied in biovars? Is the each and every isolate varied in AST profiles?
Delete line 204-206 and is not required.

Explain the reason: Interestingly, the segregation of patterns 220 based on the source of country origin of frozen shrimp revealed few inter-country resistant patterns 221 found interconnecting such as 4a, 6a, 8b, and 12a patterns, which were analogous in both the 222 countries. This influenced 44 (24.4%) isolates overlapping between China and Vietnam as shown 223 in Figure 2 and Table 3

is it the China may purchased from Vietnam?
Line 255-256: Cite the original Author reported

Line 260-261:Add a Reference

Delete:Line no 280-281: delete moreover, the study revealed that the isolates

Line 284:delete and the study revealed that
Add as with..

delete "e.g" so write such as in line no.289

Delete... issue according to the in Line no. 298-299

Delete in line no. 327 "resistant to"...one antimicrobial to resistant
Delete Line no. 329-331" on the other........ Shariff 2021)
Delete Line no 335 "and resistance"

Delete in line no.353 "and"

Additional comments

Overall improvement of the grammar is required.

---

## Round 0.2 · accepted · Accept

The authors have responded to reviewers' comments in a good manner, and the quality of the manuscript is significantly improved. The manuscript is acceptable now. However, there are still some language and experimental issues mentioned by Reviewer 1: "The written English in some sections was unclear and some incorrect words or spelling were detected". It is better for the authors to further improve the language. Thank you for your contribution to PeerJ.

Reviewer 1 ·

Basic reporting

The written English in some sections was unclear and some incorrect words or spelling were detected, but the authors have improved the English.

Mistakes related to references were corrected.

Experimental design

All the reviewers evoked the absence of genetic testing methods, and the authors responded that their project was divided into two phases, in phase they conducted isolation of E. coli and antimicrobial susceptibility testing and we generated much antimicrobial data and we decided to publish separately. In phase II they will screen all isolates for potential antibiotic resistance genes as well as molecular typing using PCR-based DNA fingerprinting such as rep-PCR and ERIC-PCR and this will be the second manuscript.

I concluded that the authors will not include the molecular results in this paper, which is in my opinion important and provided more details about the nature of resistance.

Validity of the findings

All the comments were corrected.

Reviewer 2 ·

Basic reporting

no comment

Experimental design

no comment

Validity of the findings

No comment.

·

Basic reporting

Corrected as per the suggestions

Experimental design

Corrected as per the suggestions

Validity of the findings

Corrections are included as per the suggestions

Additional comments

Corrected as per the suggestions